# Rank-Adaptive Tensor Completion Based on Tucker Decomposition

**DOI:** 10.3390/e25020225

**Published:** 2023-01-24

**Authors:** Siqi Liu, Xiaoyu Shi, Qifeng Liao

**Affiliations:** School of Information Science and Technology, ShanghaiTech University, Shanghai 201210, China

**Keywords:** tensor completion, Tucker decomposition, HOOI algorithm, rank-adaptive methods, SVT algorithm

## Abstract

Tensor completion is a fundamental tool to estimate unknown information from observed data, which is widely used in many areas, including image and video recovery, traffic data completion and the multi-input multi-output problems in information theory. Based on Tucker decomposition, this paper proposes a new algorithm to complete tensors with missing data. In decomposition-based tensor completion methods, underestimation or overestimation of tensor ranks can lead to inaccurate results. To tackle this problem, we design an alternative iterating method that breaks the original problem into several matrix completion subproblems and adaptively adjusts the multilinear rank of the model during optimization procedures. Through numerical experiments on synthetic data and authentic images, we show that the proposed method can effectively estimate the tensor ranks and predict the missing entries.

## 1. Introduction

Tensors, as a higher-order generalization of vectors and matrices, can preserve the original structure and the latent characteristics of multi-dimensional data, such as images and videos. To store and process these data, vectorization or matricization may break the adjacency relation and lead to the loss of crucial information in practical applications. Therefore, tensor analysis of data has gathered increasing attention due to its better performance in finding hidden structures and capturing potential features. Due to data missing from a transmission or insufficient collection, sometimes we have to deal with incomplete data, which increases the operation cost and the difficulty of multi-dimensional data analysis. Therefore, tensor completion, also referred to as tensor recovery, plays a key role in processing and analyzing incomplete multi-dimensional data. In tensor completion, the prediction of unknown data from a few observed entries is achieved based on the correlation between different parts of data or, simply, the hidden low-rank structure. Many real-world data lie in a latent low dimensional space since they may only relate to a small number of contributions; thus, we can use tensor completion to infer the missing data. It has already been widely applied in many scientific fields, including surrogate construction [1], image and video recovery [2,3,4], traffic data completion [5] and the MIMO (multi-input multi-output) problems in information theory [6].

The tensor completion methods can be roughly classified into model-based and non-model-based ones. Extending the idea from existing matrix completion algorithms [7,8], non-model-based methods directly formulate the tensor completion task as a rank minimization problem subject to linear constraints, and then relax it to a nuclear norm minimization problem, such as LRTC (low rank tensor completion) [9]. However, model-based methods achieve the goal differently. Based on a decomposition model with a given rank, these methods minimize the error on the known entries by adjusting the model parameters, such as CP-Wopt (CP-weighted optimization) [10], Tucker–Wopt (Tucker-weighted optimization) [5], $M^{2}\mathrm{SA}$ (multilinear subspace analysis with missing values) [11] and so on.

The computation cost of non-model-based methods is closely related to the size of the tensor, leaving little room for reducing the time complexity. On the contrary, this can be more flexibly controlled in model-based methods by altering the model size. Nonetheless, there still exists a big challenge when choosing the model’s rank. Underestimation of the tensor rank makes it harder to reach our desired accuracy, whereas overestimation may lead to a poor generalization on the unobserved part of entries. Therefore, rank-adaptive methods are needed and worth studying. In [1], a greedy scheme is designed to find the fittest rank by gradually increasing the number of factors in the CP decomposition model. In [12], from an overestimated rank, a systematic approach to reducing the CP ranks to optimal ones is developed.

Focusing on tensor completion based on Tucker decomposition, we propose a novel rank-adaptive tensor completion method and verify its efficiency through experiments on synthetic and real practical data. The rest of the paper is organized as follows. Section 2 formulates the problem setting. Section 3 presents Tucker decomposition and the related works. We propose our rank-adaptive tensor completion based on the Tucker decomposition (RATC-TD) approach in Section 4 and verify it through numerical experiments in Section 5. Finally, we conclude our work in Section 6.

## 2. Notations and the Tensor Completion Problem

Before summarizing the related work and presenting our algorithm, this section first introduces some basic definitions that will be used in the following sections, and then states our problem settings.

### 2.1. Notations

We follow the notations in [13]. Let lowercase letters denote scalars, e.g., *a*; let lowercase bold letters denote vectors, e.g., a; let bold capital letter denote matrices, e.g., U; let capital fraktur letter denote high-order tensors, e.g., T. The *i*th element of a vector a is denoted as $a_{i}$; the element of matrix U with index (i,j) is denoted by $u_{ij}$; the element of *N*-way tensor T with index $\left(i_{1},i_{2},\cdots ,i_{N}\right)$ is denoted by $t_{i_{1}i_{2}\cdots i_{N}}$. A *tensor* can be viewed as a multi-dimensional array, and the number of dimensions is generally called the *order* of the tensor. For example, matrices are second-order tensors, while vectors are first-order tensors. A *mode-n fiber* is a vector extracted from a tensor by fixing all indices except the *n*th one. Consider a matrix as an example; mode-1 fibers refer to the columns, and mode-2 fibers refer to the rows. The *mode-n unfolding* of a tensor $T\in R^{I_{1}\times I_{2}\times \cdots \times \;I_{N}}$ is to rearrange the mode-*n* fibers as the columns of the resulting matrix, denoted as $T_{\left(n\right)}\in R^{I_{n}\times I_{1}\cdots I_{n-1}I_{n+1}\cdots I_{N}}$. Naturally, the reverse of *unfolding*, the process of rearranging $T_{\left(n\right)}\in R^{I_{n}\times I_{1}\cdots I_{n-1}I_{n+1}\cdots I_{N}}$ into $T\in R^{I_{1}\times I_{2}\times \cdots \times \;I_{N}}$, is called *folding*.

The *n-mode matrix product* is the multiplication of a tensor $T\in R^{I_{1}\times I_{2}\times \cdots \times \;I_{N}}$ with a matrix $U\in R^{J\times I_{n}}$, denoted as $X=T\times_{n}U$, where $X\in R^{I_{1}\times \cdots \times I_{n-1}\times J\times I_{n+1}\times \cdots \times I_{N}}$. It can be regarded as multiplying each mode-*n* fiber of T with U. The matricized form is given by
(1)$$X_{\left(n\right)}=UT_{\left(n\right)},$$
and the element-wise operations can be obtained by
(2)$$x_{i_{1}\cdots i_{n-1}ji_{n+1}\cdots i_{N}}=\sum_{i_{n}=1}^{I_{n}}t_{i_{1}\cdots i_{n-1}i_{n}i_{n+1}\cdots i_{N}}u_{ji_{n}}.$$

The *norm* of a tensor $T\in R^{I_{1}\times I_{2}\times \cdots \times \;I_{N}}$ is a higher-order analog of the matrix Frobenius norm, defined by
(3)$${∥T∥}_{F}=\sqrt{\sum_{i_{1}=1}^{I_{1}}\sum_{i_{2}=1}^{I_{2}}\cdots \sum_{i_{N}=1}^{I_{N}}t_{i_{1}i_{2}\cdots i_{N}}^{2}}.$$

The *inner product* of two same sized tensors $T,X\in R^{I_{1}\times I_{2}\times \cdots \times \;I_{N}}$ is defined by
(4)$$\langle T,X\rangle =\sum_{i_{1}=1}^{I_{1}}\sum_{i_{2}=1}^{I_{2}}\cdots \sum_{i_{N}=1}^{I_{N}}t_{i_{1}i_{2}\cdots i_{N}}x_{i_{1}i_{2}\cdots i_{N}}.$$

### 2.2. The Tensor Completion Problem

Tensor completion is the problem of predicting missing entries through a few sampled entries of a tensor based on the assumption that it has a low-rank structure. Supposing we have a partially observed tensor $T\in R^{I_{1}\times \cdots \times I_{N}}$, the set of indices where entries are observed is denoted as Ω. Let $P_{\Omega }$ be an operator on $R^{I_{1}\times \cdots \times I_{N}}$, such that $P_{\Omega }\left(X\right)$ is equal to X on Ω and is equal to 0 otherwise. The low-rank tensor completion problem can be formulated as
(5)$$\begin{array}{cc} \min_{X}\; & \mathrm{rank}\;\left(X\right)\;s.t.\;P_{\Omega }\left(X\right)=P_{\Omega }\left(T\right). \end{array}$$

Although the formulation seems easy, how to measure the rank of a tensor is still an open question. In previous research, the CP rank [1,12] and the Tucker rank [9,14,15] were mostly considered. After the tensor-train (TT) decomposition was proposed in [16], tensor completion based on the TT rank was studied in [3,17,18]. The low-tubal rank tensor completion problem was studied in [19,20,21]. In addition, the combination of the CP and the Tucker ranks was investigated in [22].

Since matrices are second-order tensors, matrix completion can be viewed as a special case of tensor completion. Using the matrix nuclear norm to approximate the rank of a matrix is studied for matrix completion in [23]. Liu et al. [9] generalized this to the tensor scheme by defining the nuclear norm of tensors and proposing the following low-rank tensor completion problem:(6)$$\begin{array}{cc} \min_{X}\; & {∥X∥}_{*}\;s.t.\;P_{\Omega }\left(X\right)=P_{\Omega }\left(T\right). \end{array}$$
The nuclear norm of a tensor is defined by ${∥X∥}_{*}:=\sum_{n=1}^{N}{∥X_{\left(n\right)}∥}_{*}$, where $∥X_{\left(n\right)}{∥}_{*}$ is the nuclear norm of the matrix $X_{\left(n\right)}$. The nuclear norm can be computed by the sum of the singular values. However, the computation involves SVD (singular value decomposition) of the unfolding matrix $T_{\left(n\right)}$ in each iteration, which is expensive when the size of T is large. To control the computation cost, this problem is reformulated based on tensor decomposition [5,11]. Particularly, the low-rank tensor completion problem based on Tucker decomposition can be written as
(7)$$\begin{array}{cc} \min_{G,A_{1}\cdots ,A_{N}}\; & {∥T-X∥}_{\Omega }^{2}\;\\ s.t.\;X=G\times_{1}A_{1}\cdots \times_{N}A_{N},\; & A_{n}^{T}A_{n}=I^{R_{n}\times R_{n}}\;,\;n=1,\cdots ,N, \end{array}$$
where X is the completed goal tensor, $\left(R_{1},\cdots ,R_{N}\right)$ is a predefined multilinear rank, $A_{n}\in R^{I_{n}\times R_{n}}$ and ${∥\cdot ∥}_{\Omega }$ denote the norm on the observation elements, i.e., $∥P_{\Omega }{\left(\cdot \right)∥}_{F}$. Methods are actively developed to solve this problem, including $M^{2}\mathrm{SA}$ [11], gHOI (generalized higher-order orthogonal iteration) [24] and Tucker–Wopt [5]. However, these algorithms typically require the multilinear ranks given a priori, both underestimating and overestimating the ranks can result in inaccurate completion results. To handle this issue, we in this work propose a rank-adaptive tensor completion approach without requiring the multilinear ranks given a priori.

## 3. The Tucker Decomposition and Its Computation

We review Tucker decomposition and its related algorithms in this section.

### 3.1. Tucker Decomposition

Tucker decomposition was first introduced by Tucker [25]. For a *N*th-order tensor $T\in R^{I_{1}\times I_{2}\times \cdots \times I_{N}}$, its Tucker decomposition has the form
(8)$$\begin{array}{c} T=G\times_{1}A_{1}\times_{2}A_{2}\cdots \times_{N}A_{N}, \end{array}$$
where $G\in R^{R_{1}\times \cdots \times R_{N}}$ is a core tensor and $A_{n}\in R^{I_{n}\times R_{n}}$, n=1,2,⋯,N, are factor matrices. For simplicity, we assume all the factor matrices have orthogonal columns. $R_{n}$ is the rank of the mode-*n* unfolding matrix $T_{\left(n\right)}$. As described in [13], Tucker decomposition can be viewed as a higher-order generalization of PCA (principal component analysis) in some sense. In Tucker decomposition, the columns of the factor matrix represent the components in each mode, and each element of the core tensor characterizes the level of interaction between the components in different modes.

The tuple $\left(R_{1},\cdots ,R_{N}\right)$ is called Tucker rank or multilinear rank. However, practically, we prefer to use the truncated version of Tucker decomposition, where we set $R_{n}<\mathrm{rank}\left(T_{\left(n\right)}\right)$ for one or more *n*. Given the truncation $\left(R_{1},\cdots ,R_{N}\right)$, the approximation of the truncated Tucker decomposition of a tensor $T\in R^{I_{1}\times \cdots \times I_{N}}$ can be described as
(9)$$\begin{array}{cc} \min_{G,A_{1}\cdots ,A_{N}}\; & {∥T-X∥}_{F}^{2}\;\\ s.t.\;X=G\times_{1}A_{1}\cdots \times_{N}A_{N},\; & A_{n}^{T}A_{n}=I^{R_{n}\times R_{n}}\;,\;n=1,\cdots ,N, \end{array}$$
where $G\in R^{R_{1}\times \cdots \times R_{N}}$, $A_{n}\in R^{I_{n}\times R_{n}}$ is the *n*th factor matrix, and $I^{R_{n}\times R_{n}}$ is an identity matrix of size $R_{n}\times R_{n}$.

### 3.2. The Higher-Order Orthogonal Iteration (HOOI) Algorithm

There are several approaches to computing the truncated Tucker decomposition of a tensor. One of the most popular methods is the higher-order orthogonal iteration (HOOI) algorithm [26], also referred to as the alternative least square (ALS) algorithm for Tucker decomposition. It is an alternative iterating method, where we fix all except one factor matrix each time, and then minimize the objective function in (Equation 9). Specifically, given initial guesses $\left\{A_{n}^{\left(0\right)}:n=1,\cdots ,N\right\}$, at the *k*th iteration, for each n=1,⋯,N, we fix all the factor matrices except $A_{n}$, and then find the optimal solution to the subproblem
(10)$$\begin{array}{cc} \min_{G,A_{n}^{\left(k\right)}}\;{∥T-X∥}_{F}^{2}\\ s.t.\;X=G\times_{1}A_{1}^{\left(k\right)}\cdots \times_{n}A_{n}^{\left(k\right)}\times_{n+1}A_{n+1}^{\left(k-1\right)}\cdots & \times_{N}A_{N}^{\left(k-1\right)},\;{\left(A_{n}^{\left(k\right)}\right)}^{T}A_{n}^{\left(k\right)}=I^{R_{n}\times R_{n}}\;, \end{array}$$
where $G\in R^{R_{1}\times \cdots \times R_{N}}$, $A_{n}^{\left(k\right)}\in R^{I_{n}\times R_{n}}$. Denoting $B:=G\times_{n}A_{n}^{\left(k\right)}$, this optimization problem can be simplified as
(11)$$\begin{array}{cc} \min_{B}\;{∥T-X∥}_{F}^{2}\;\\ s.t.\;X=B\times_{1}A_{1}^{\left(k\right)}\cdots \times_{n-1}A_{n-1}^{\left(k\right)}\times_{n+1}A_{n+1}^{\left(k-1\right)}\cdots & \times_{N}A_{N}^{\left(k-1\right)},\;\mathrm{rank}\left(B_{\left(n\right)}\right)\leq R_{n}, \end{array}$$
where $B\in R^{R_{1}\times \cdots R_{n-1}\times I_{n}\times R_{n+1}\cdots R_{N}}$ and $B_{\left(n\right)}$ is the mode-*n* unfolding matrix form of B. It can be considered as a constrained least squares problem [27], and thus the solution to (Equation 10) can be easily obtained by the following steps:Compute $B^{*}=T\times_{1}{\left(A_{1}^{\left(k\right)}\right)}^{T}\cdots \times_{n-1}{\left(A_{n-1}^{\left(k\right)}\right)}^{T}\times_{n+1}{\left(A_{n+1}^{\left(k-1\right)}\right)}^{T}\cdots \times_{N}{\left(A_{N}^{\left(k-1\right)}\right)}^{T}$;Unfold $B^{*}$ at the *n*th mode to obtain $B_{\left(n\right)}^{*}$, then perform a truncated rank-$R_{n}$ singular value decomposition ${\left(B_{\left(n\right)}^{*}\right)}_{\left[R\right]}=U\Sigma V^{T}=B_{\left(n\right)}$;Compute $A_{n}^{\left(k\right)}=U$ and $G_{\left(n\right)}=\Sigma V^{T}$.

We iteratively solve these subproblems until a given stopping criterion is reached, i.e., the difference between the solutions at two adjacent iterations is small enough, or the value of the objective function decreases very slightly. The complete procedure of HOOI is presented in Algorithm 1.
**Algorithm 1** The high-order orthogonal iteration (HOOI) algorithm [26]**Input:** Tensor $T\in R^{I_{1}\times I_{2}\times \cdots \times I_{N}}$ and truncation $\left(R_{1},R_{2},\cdots ,R_{N}\right)$.**Output:** Core tensor $G\in R^{R_{1}\times \cdots \times R_{N}}$, and factor matrices $A_{n}\in R^{I_{n}\times R_{n}}$ for n=1,2,⋯,N.1:Initialize $A_{n}^{\left(0\right)}\in R^{I_{n}\times R_{n}}$ for n=1,2,⋯,N using HOSVD.2:k←0.3:**while** not converge **do**4:    k←k+1.5:    **for** n=1,2,⋯,N **do**6:        $B\leftarrow T\times_{1}{\left(A_{1}^{\left(k\right)}\right)}^{T}\cdots \times_{n-1}{\left(A_{n-1}^{\left(k\right)}\right)}^{T}\times_{n+1}{\left(A_{n+1}^{\left(k-1\right)}\right)}^{T}\cdots \times_{N}{\left(A_{N}^{\left(k-1\right)}\right)}^{T}$.7:        $B_{\left(n\right)}\leftarrow$ mode-*n* unfolding matrix of B.8:        $U,\Sigma ,V^{T}\leftarrow$ truncated rank-$R_{n}$ SVD of $B_{\left(n\right)}$.9:        $A_{n}^{\left(k\right)}\leftarrow U$.10:    **end for**11:**end while**12:$G\leftarrow \Sigma V^{T}$ folding at mode-*n*.

### 3.3. The Rank-Adaptive HOOI Algorithm

HOOI requires the multilinear rank $\left(R_{1},\cdots ,R_{N}\right)$ given a priori, which is hard to be determined in practice. Instead of (Equation 9), Xiao and Yang [27] consider the following form of the low multilinear rank approximation problem:(12)$$\begin{array}{c} \min_{G,A_{1}\cdots ,A_{N}}\;\left(R_{1},R_{2},\cdots ,R_{N}\right){\;s.t.\;∥X-T∥}_{F}^{2}<\epsilon {∥T∥}_{F}^{2},\\ X=G\times_{1}A_{1}\cdots \times_{N}A_{N},\;A_{n}^{T}A_{n}=I^{R_{n}\times R_{n}}\;,\;n=1,\cdots ,N, \end{array}$$
where ϵ is a given tolerance. A rank-adaptive HOOI algorithm is proposed in [27], which adjusts the truncation $R_{n}$ for dimension *n* in the HOOI iterations by
(13)$$\begin{array}{c} R_{n}^{\left(k\right)}=\underset{R}{\mathrm{argmin}}\;∥B_{\left(n\right)}-{\left(B_{\left(n\right)}\right)}_{\left[R\right]}{∥}_{F}^{2}\leq {∥B∥}_{F}^{2}-\left(1-\epsilon \right){∥T∥}_{F}^{2}, \end{array}$$
where ${\left(B_{\left(n\right)}\right)}_{\left[R\right]}$ is the best rank-*R* approximation of $B_{\left(n\right)}$. For the full SVD of $B_{\left(n\right)}=U\Sigma V^{T}$, it can be calculated by ${\left(B_{\left(n\right)}\right)}_{\left[R\right]}=U_{:,1:R}\Sigma_{1:R,1:R}V_{:,1:R}^{T}$. In [27], it is proven that (Equation 13) is a local optimal strategy for updating $R_{n}$, i.e., the optimal solution of the following problem:(14)$$\begin{array}{c} \min_{B}\;\mathrm{rank}\left(B_{\left(n\right)}\right){\;s.t.\;∥X-T∥}_{F}^{2}<\epsilon {∥T∥}_{F}^{2},\\ X=B\times_{1}A_{1}^{\left(k\right)}\cdots \times_{n-1}A_{n-1}^{\left(k\right)}\times_{n+1}A_{n+1}^{\left(k-1\right)}\cdots \times_{N}A_{N}^{\left(k-1\right)}. \end{array}$$

Details of the above procedure are summarized in Algorithm 2.
**Algorithm 2** The rank-adaptive HOOI algorithm [27]**Input:** Tensor $T\in R^{I_{1}\times I_{2}\times \cdots \times I_{N}}$, error tolerance ϵ, initial guess of factor matrices $A_{n}^{\left(0\right)}\in R^{I_{n}\times R_{n}^{\left(0\right)}}$ for n=1,2,⋯,N and initial truncation $\left(R_{1}^{\left(0\right)},R_{2}^{\left(0\right)},\cdots ,R_{N}^{\left(0\right)}\right)$.**Output:** Truncation $\left(R_{1}^{\left(k\right)},R_{2}^{\left(k\right)},\cdots ,R_{N}^{\left(k\right)}\right)$, core tensor $G\in R^{R_{1}^{\left(k\right)}\times \cdots \times R_{N}^{\left(k\right)}}$ and factor matrices $A_{n}^{\left(k\right)}\in R^{I_{n}\times R_{n}^{\left(k\right)}}$ for n=1,2,⋯,N.1:$G^{\left(0\right)}\leftarrow T\times_{1}{\left(A_{1}^{\left(0\right)}\right)}^{T}\cdots \times_{N}{\left(A_{N}^{\left(0\right)}\right)}^{T}$.2:k←0.3:**while** not converge **do**4:    k←k+1.5:    **for all** n∈{1,2,⋯,N} **do**6:        $B\leftarrow T\times_{1}{\left(A_{1}^{\left(k\right)}\right)}^{T}\cdots \times_{n-1}{\left(A_{n-1}^{\left(k\right)}\right)}^{T}\times_{n+1}{\left(A_{n+1}^{\left(k-1\right)}\right)}^{T}\cdots \times_{N}{\left(A_{N}^{\left(k-1\right)}\right)}^{T}$.7:        $B_{\left(n\right)}\leftarrow$ mode-*n* unfolding matrix of B.8:        $U,\Sigma ,V^{T}\leftarrow$ full SVD of $B_{\left(n\right)}$.9:        $R_{n}^{\left(k\right)}\leftarrow$ minimum *R* such that $\sum_{r>R}\Sigma_{r,r}^{2}{<∥B∥}_{F}^{2}-\left(1-\epsilon \right){∥T∥}_{F}^{2}$.10:        $A_{n}^{\left(k\right)}\leftarrow U_{:,1:R_{n}^{\left(k\right)}}$.11:    **end for**12:    $G\leftarrow \Sigma_{1:R_{n}^{\left(k\right)},1:R_{n}^{\left(k\right)}}V_{:,1:R_{n}^{\left(k\right)}}^{T}$ folding at mode-*n*.13:**end while**

## 4. A Rank-Adaptive Tensor Recovery Scheme

This section proposes a rank-adaptive tensor completion scheme based on truncated Tucker decomposition (RATC-TD).

Analogous to the rank-adaptive HOOI algorithm, instead of (Equation 7), we consider a different form of the low-rank tensor completion problem:(15)$$\begin{array}{c} \min_{G,A_{1}\cdots ,A_{N}}\;\left(R_{1},R_{2},\cdots ,R_{N}\right)\;s.t.\;P_{\Omega }\left(X\right)=P_{\Omega }\left(T\right),\\ X=G\times_{1}A_{1}\cdots \times_{N}A_{N},\;A_{n}^{T}A_{n}=I^{R_{n}\times R_{n}}\;,\;n=1,\cdots ,N. \end{array}$$

Note that if the data are noisy, the constraint must be relaxed to ${∥X-T∥}_{\Omega }^{2}<\epsilon {∥T∥}_{\Omega }^{2}$, where ϵ is a given tolerance of the relative error on the observation part between the original tensor and the completed one.

Using an alternative optimization technique, problem (Equation 15) can be divided into *N* subproblems, which will be detailedly represented in Section 4.1. To address those subproblems, the singular value thresholding (SVT) algorithm [7] is introduced in Section 4.2. The entire algorithm is summarized in Section 4.3.

### 4.1. Alternative Optimization in Tensor Completion

We solve (Equation 15) by an alternative iteration method. For each n=1,⋯,N, initialize $R_{n}^{\left(0\right)}=I_{n}$, and the initial guess $A_{n}^{\left(0\right)}$ can be either randomly set or given by HOSVD (higher-order SVD) [13] of $P_{\Omega }\left(T\right)$. At the *k*th iteration, for each n=1,⋯,N, by fixing all the factor matrices except $A_{n}$, we solve the subproblem
(16)$$\begin{array}{cc} \min_{G,A_{n}^{\left(k\right)}}\;\left(R_{n}^{\left(k\right)}\right)\; & s.t.\;P_{\Omega }\left(X\right)=P_{\Omega }\left(T\right),\\ X=G\times_{1}A_{1}^{\left(k\right)}\cdots \times_{n}A_{n}^{\left(k\right)}\times_{n+1}A_{n+1}^{\left(k-1\right)} & \cdots \times_{N}A_{N}^{\left(k-1\right)},\;{\left(A_{n}^{\left(k\right)}\right)}^{T}A_{n}^{\left(k\right)}=I^{R_{n}\times R_{n}}. \end{array}$$

Denoting $B:=G\times_{n}A_{n}^{\left(k\right)}$, (Equation 16) is equivalent to
(17)$$\begin{array}{c} \min_{B}\;\mathrm{rank}\left(B_{\left(n\right)}\right)\;s.t.\;P_{\Omega }\left(X\right)=P_{\Omega }\left(T\right),\\ X=B\times_{1}A_{1}^{\left(k\right)}\cdots \times_{n-1}A_{n-1}^{\left(k\right)}\times_{n+1}A_{n+1}^{\left(k-1\right)}\cdots \times_{N}A_{N}^{\left(k-1\right)}, \end{array}$$
where $B_{\left(n\right)}$ is the mode-*n* unfolding matrix form of B. However, this rank minimization problem is NP-hard. Its tightest convex relaxation is
(18)$$\begin{array}{c} \min_{B}\;{∥B_{\left(n\right)}∥}_{*}\;s.t.\;P_{\Omega }\left(X\right)=P_{\Omega }\left(T\right),\\ X=B\times_{1}A_{1}^{\left(k\right)}\cdots \times_{n-1}A_{n-1}^{\left(k\right)}\times_{n+1}A_{n+1}^{\left(k-1\right)}\cdots \times_{N}A_{N}^{\left(k-1\right)}, \end{array}$$
where $∥B_{\left(n\right)}{∥}_{*}$ is the nuclear norm of the matrix $B_{\left(n\right)}$, which can be computed by the sum of the singular values. The matrix form of this problem can be written as [27]
(19)$$\begin{array}{c} \min_{B}\;{∥B∥}_{*}\;s.t.\;P_{\Omega }\left(BM^{T}\right)=P_{\Omega }\left(T_{\left(n\right)}\right),\\ M=A_{N}\cdots ⊗A_{n+1}⊗A_{n-1}\cdots ⊗A_{1}, \end{array}$$
where $T_{\left(n\right)}$ is the mode-*n* unfolding matrix form of T, and ⊗ is the Kronecker product. In this work, we use the singular value thresholding (SVT) algorithm [7] to obtain a proximity solution to (Equation 19).

### 4.2. Solving the Subproblems Using SVT

Following the notation used in [7], we first introduce the singular value thresholding operator. Consider the skinny singular value decomposition of a matrix $X\in R^{n_{1}\times n_{2}}$ of rank *r*,
(20)$$X=U\Sigma V^{T},\;\Sigma =\mathrm{diag}\left({\left\{\sigma_{i}\right\}}_{1\leq i\leq r}\right),$$
where U and V are, respectively, $n_{1}\times r$ and $n_{2}\times r$ matrices with orthonormal columns, and the singular values $\sigma_{i}>0$ for i=1,⋯,r. Given the shrinkage threshold τ>0, the singular value thresholding operator $D_{\tau }$ is defined by
(21)$$D_{\tau }\left(X\right):=UD_{\tau }\left(\Sigma \right)V^{T},\;D_{\tau }\left(\Sigma \right)=\mathrm{diag}\left({\left\{\max\left(\sigma_{i}-\tau ,0\right)\right\}}_{1\leq i\leq r}\right).$$

According to the deduction of [7], for each τ>0 and $Y\in R^{n_{1}\times n_{2}}$, the singular value shrinkage operator (Equation 21) obeys
(22)$$D_{\tau }\left(Y\right)=\underset{X}{\mathrm{argmin}}\;\left\{{\tau ∥X∥}_{*}+\frac{1}{2}{∥X-Y∥}_{F}^{2}\right\}.$$

The SVT algorithm utilizes the singular value thresholding operator $D_{\tau }$ and its property (Equation 22) to handle the subproblem (Equation 19). Consider the proximal problem for τ>0,
(23)$$\begin{array}{c} \min_{B}{\;\tau ∥B∥}_{*}+\frac{1}{2}{∥B∥}_{F}^{2}\\ s.t.\;P_{\Omega }\left(BM^{T}\right)=P_{\Omega }\left(T_{\left(n\right)}\right), \end{array}$$
the solution of which converges to that of (Equation 19) as τ→∞. This optimization problem can be solved by a Lagrangian multiplier method known as the Uzawa’s algorithm [28].

Given the Lagrangian of (Equation 23)
(24)$$\begin{array}{c} L\left(B,Y\right)={\tau ∥B∥}_{*}+\frac{1}{2}{∥B∥}_{F}^{2}+\langle Y,P_{\Omega }\left(T_{\left(n\right)}-BM^{T}\right)\rangle , \end{array}$$
where Y has the same size as $T_{\left(n\right)}$, the optimal $B^{*}$ and $Y^{*}$ should satisfy
(25)$$L\left(B^{*},Y^{*}\right)=\underset{B}{\inf}\;\underset{Y}{\sup}\;L\left(B,Y\right)=\underset{Y}{\sup}\;\underset{B}{\inf}\;L\left(B,Y\right).$$

Starting with $Y^{\left(0\right)}=0$, Uzawa’s algorithm finds the saddle point $\left(B^{*},Y^{*}\right)$ through an iterative procedure given by
(26)$$\left\{\begin{array}{c} B^{\left(k\right)}={\mathrm{argmin}}_{B}\;L\left(B,Y^{\left(k-1\right)}\right)\\ Y^{\left(k\right)}=Y^{\left(k-1\right)}+\delta_{k}P_{\Omega }\left(T_{\left(n\right)}-B^{\left(k\right)}M^{T}\right), \end{array}\right.$$
where ${\left\{\delta_{k}\right\}}_{k\geq 1}>0$ are scalar step sizes. The sequence $\left\{B^{\left(k\right)}\right\}$ converges to the unique solution to (Equation 23). The update of Y is actually based on the gradient descent method if we note that
(27)$$\partial_{Y}L\left(B,Y\right)=P_{\Omega }\left(T_{\left(n\right)}-BM^{T}\right).$$

Now, we have to compute the minimizer of (Equation 26). Observe that the factor matrices $\left\{A_{k}:k=1,\cdots ,N\right\}$ have orthogonal columns. Based on the orthogonal invariance property of the Frobenius norm, we have
(28)$${∥B∥}_{F}^{2}={∥B\times_{1}A_{1}^{\left(k\right)}\cdots \times_{n-1}A_{n-1}^{\left(k\right)}\times_{n+1}A_{n+1}^{\left(k-1\right)}\cdots \times_{N}A_{N}^{\left(k-1\right)}∥}_{F}^{2},$$
which in the matrix form gives that
(29)$${∥B∥}_{F}^{2}={∥BM^{T}∥}_{F}^{2}.$$

Utilizing this property,
(30)$$\begin{array}{cc} \; & {\mathrm{argmin}}_{B}{\;\tau ∥B∥}_{*}+\frac{1}{2}{∥B∥}_{F}^{2}+\langle Y,P_{\Omega }\left(T_{\left(n\right)}-BM^{T}\right)\rangle \\ =\; & {\mathrm{argmin}}_{B}{\;\tau ∥B∥}_{*}+\frac{1}{2}{∥BM^{T}∥}_{F}^{2}+\langle Y,P_{\Omega }\left(T_{\left(n\right)}-BM^{T}\right)\rangle \\ =\; & {\mathrm{argmin}}_{B}{\;\tau ∥B∥}_{*}+\frac{1}{2}{∥BM^{T}-P_{\Omega }\left(Y\right)∥}_{F}^{2}\\ =\; & {\mathrm{argmin}}_{B}{\;\tau ∥B∥}_{*}+\frac{1}{2}{∥B-P_{\Omega }\left(Y\right)M∥}_{F}^{2}. \end{array}$$

According to (Equation 22), the optimal $B^{*}$ is given by $D_{\tau }\left(P_{\Omega }\left(Y\right)M\right)=D_{\tau }\left(YM\right)$ since $P_{\Omega }\left(Y\right)=Y$ for all k≥0. Therefore, Uzawa’s algorithm finally takes the form
(31)$$\left\{\begin{array}{c} B^{\left(k\right)}=D_{\tau }\left(Y^{\left(k-1\right)}M\right)\\ Y^{\left(k\right)}=Y^{\left(k-1\right)}+\delta_{k}P_{\Omega }\left(T_{\left(n\right)}-B^{\left(k\right)}M^{T}\right), \end{array}\right.$$
also referred to as the shrinkage iterations in SVT.

To obtain an approximation to the solution of (Equation 19), we choose a large enough τ and perform the iterations (Equation 31) until the stopping criteria $∥T-B^{\left(k\right)}M^{T}{∥}_{\Omega }^{2}<\epsilon {∥T∥}_{\Omega }^{2}$ are reached, starting with $Y^{\left(0\right)}=0\in R^{I_{1}\times \cdots \times I_{N}}$.

The overall process of solving the subproblem (Equation 18) is shown in Algorithm 3.
**Algorithm 3** The SVT algorithm for solving (Equation 18)**Input:** Set of observed indices Ω, tensor with observed entries $P_{\Omega }\left(T\right)\in R^{I_{1}\times I_{2}\times \cdots \times I_{N}}$, fixed factor matrices $A_{m}\in R^{I_{m}\times R_{m}}$ for m={1,2,⋯,N}/n, error tolerance ϵ, shrinkage threshold τ and scalar step sizes ${\left\{\delta_{k}\right\}}_{k\geq 1}$.**Output:** Optimized B.1:$P_{\Omega }\left(T\right)\leftarrow$ mode-*n* unfolding matrix of $P_{\Omega }\left(T\right)$.2:$M\leftarrow A_{N}\cdots ⊗A_{n+1}⊗A_{n-1}\cdots ⊗A_{1}$.3:Initialize $B^{\left(0\right)}\leftarrow \left(P_{\Omega }\left(T\right)\right)M$,   $Y^{\left(0\right)}\leftarrow 0$.4:k←0.5:**while**$∥T-B^{\left(k\right)}M^{T}{∥}_{\Omega }^{2}\geq \epsilon {∥T∥}_{\Omega }^{2}$**do**6:    k←k+1.7:    Update $B^{\left(k\right)}\leftarrow D_{\tau }\left(Y^{\left(k-1\right)}M\right)$, where $D_{\tau }$ is defined in (Equation 21).8:    Update $Y^{\left(k\right)}\leftarrow Y^{\left(k-1\right)}+\delta_{k}P_{\Omega }\left(T_{\left(n\right)}-B^{\left(k\right)}M^{T}\right)$.9:**end while**10:$B\leftarrow B^{\left(k\right)}$ folding at mode-*n*.

### 4.3. The Rank-Adaptive Tensor Completion Algorithm

This section summarizes our proposed method, namely the rank-adaptive tensor completion algorithm. Based on Tucker decomposition, this algorithm can complete a tensor with only a few observed entries and adaptively estimate its multilinear rank.

Given a tensor $T\in R^{I_{1}\times \cdots \times I_{N}}$, only entries with indices in the set Ω are observed. Our goal is to predict the unobserved entries based on the premise that T has a low-rank structure. In this work, we consider the multilinear rank of the tensor. We solve this problem by finding a low multilinear rank tensor X whose entries on the observation part satisfy ${∥X-T∥}_{\Omega }^{2}<\epsilon {∥T∥}_{\Omega }^{2}$, where ϵ is a given tolerance. Here, we restate our problem setting (Equation 15)
$$\begin{array}{c} \min_{G,A_{1}\cdots ,A_{N}}\;\left(R_{1},R_{2},\cdots ,R_{N}\right)\;s.t.\;P_{\Omega }\left(X\right)=P_{\Omega }\left(T\right),\\ X=G\times_{1}A_{1}\cdots \times_{N}A_{N},\;A_{n}^{T}A_{n}=I^{R_{n}\times R_{n}}\;,\;n=1,\cdots ,N. \end{array}$$

Similar to HOOI, we solve (Equation 15) by alternative iterations. By fixing all the factor matrices except one, we obtain the rank minimization subproblem (Equation 17). Since it is NP-hard, we instead consider its convex relaxation form (Equation 18), i.e., minimizing the nuclear norm, which can be handled by the SVT algorithm. The solution can be computed using Algorithm 3. After optimized B in (Equation 18) is obtained, we update $R_{n}^{\left(k\right)}=\min\left(R_{n}^{\left(k-1\right)},\;\mathrm{rank}\left(B_{\left(n\right)}\right)\right)$ to ensure that $R_{n}^{\left(k\right)}$ is monotone decreasing. We perform the iterations until some stopping criteria are satisfied, e.g., the estimated rank $\left(R_{1},R_{2},\cdots ,R_{N}\right)$ remains unchanged, and the factor matrices have slight improvement, or the maximum number of iterations is reached. We present the detailed procedure of our proposed method in Algorithm 4.
**Algorithm 4** The rank-adaptive tensor completion based on Tucker decomposition (RATC-TD) algorithm**Input:** Set of observed indices Ω, tensor with observed entries $P_{\Omega }\left(T\right)\in R^{I_{1}\times I_{2}\times \cdots \times I_{N}}$ and initial guess of factor matrices $A_{n}^{\left(0\right)}\in R^{I_{n}\times R_{n}^{\left(0\right)}}$ for n=1,2,⋯,N.**Output:** Estimated rank $\left(R_{1}^{\left(k\right)},R_{2}^{\left(k\right)},\cdots ,R_{N}^{\left(k\right)}\right)$, core tensor $G\in R^{R_{1}^{\left(k\right)}\times \cdots \times R_{N}^{\left(k\right)}}$ and factor matrices $A_{n}^{\left(k\right)}\in R^{I_{n}\times R_{n}^{\left(k\right)}}$ for n=1,2,⋯,N.1:k←0.2:Initialize $\left(R_{1}^{\left(0\right)},R_{2}^{\left(0\right)},\cdots ,R_{N}^{\left(0\right)}\right)\leftarrow \left(I_{1},I_{2},\cdots ,I_{N}\right)$.3:**while** not converge **do**4:    **for all** n=1,2,⋯,N **do**5:        k←k+1.6:        B← solution to (Equation 18) using Algorithm 3.7:        $B_{\left(n\right)}\leftarrow$ mode-*n* unfolding matrix of B.8:        $U,\Sigma ,V^{T}\leftarrow$ full-SVD of $B_{\left(n\right)}$.9:        $R_{n}^{\left(k\right)}\leftarrow \min\left(R_{n}^{\left(k-1\right)},\;\mathrm{rank}\left(B_{\left(n\right)}\right)\right)$.10:        $A_{n}^{\left(k\right)}\leftarrow U_{:,1:R_{n}^{\left(k\right)}}$.11:    **end for**12:    $G\leftarrow \Sigma_{1:R_{n}^{\left(k\right)},1:R_{n}^{\left(k\right)}}V_{:,1:R_{n}^{\left(k\right)}}^{T}$ folding at mode-*n*.13:**end while**

## 5. Numerical Experiments

In this section, numerical experiments are conducted to the effectiveness of our proposed rank-adaptive tensor completion based on Tucker decomposition (RATC-TD) algorithm.

### 5.1. Test Problem 1: The Recovery of Third-Order Tensors

Many tensor recovery algorithms in the Tucker scheme require a priori given the rank of the tensor, but our method does not require this. In this test problem, synthetic data are considered, and we set the random composition tensor in the same way as [9]. We consider the tensor $T\in R^{I_{1}\times I_{2}\times I_{3}}$ and obtain tensors by multiplying a randomly generated kernel G of size $r_{1}\times r_{2}\times r_{3}$ with randomly generated factor matrices $A_{i}$, where $A_{i}\in R^{I_{i}\times r_{i}},i=1,2,3$. So, the tensor T can be represented by $G\times_{1}A_{1}\times_{2}A_{2}\times_{3}A_{3}$, and the number of entries of T is denoted by |T|. By setting tensors in this way, we can obtain random tensors and ensure that the obtained tensors have a low-rank structure, which can be handled well by the classical low-rank tensor recovery methods.

In this experiment, we set the tensor size as 50×50×50, the kernel size as 5×5×5, and the kernel data are randomly generated from the interval [0,1]. The factor matrices are of size 50×5, and the data of factor matrices are randomly generated from [−0.5,0.5]. In order to show the robustness and reliability of our proposed algorithm, we add a small perturbation to the synthetic data. We add Gaussian noise with zero mean and standard deviation as 0.1 times the element mean and take the tensor with Gaussian noise as the ground truth data. As described above, our algorithm does not require the rank in advance, and the algorithm takes the size of the tensor as the initial tensor rank. With the continuous operation of the algorithm, the tucker rank is constantly reduced, and the exact rank can be approximated.

We compare the proposed algorithm with other Tucker-decomposition-based tensor completion algorithms (that require appropriate ranks given a priori), including gHOI [24], M${}^{2}$SA [11] and Tucker–Wopt [5], and test the sample estimation error $error_{obs}$ and out-of-sample estimation error $error_{val}$ with different initial ranks at sampling rates *r* (the proportion of known data in the total data) of 0.05, 0.1 and 0.2, respectively. The size of the observed index set Ω is r|T|, and each index in Ω is randomly generated through the uniform distribution. $\bar{\Omega }$ denotes the complementary set of Ω. The specific definitions of two kinds of errors are given by the following formulas: (32)$$\begin{array}{c} error_{obs}=\frac{∥P_{\Omega }{\left(X-T\right)∥}_{F}^{2}}{∥P_{\Omega }{\left(T\right)∥}_{F}^{2}}, \end{array}$$(33)$$\begin{array}{c} error_{val}=\frac{∥P_{\bar{\Omega }}{\left(X-T\right)∥}_{F}^{2}}{∥P_{\bar{\Omega }}{\left(T\right)∥}_{F}^{2}}, \end{array}$$
where X is the result obtained by completion algorithms.

To ensure the generalization of the proposed algorithms, we set the error tolerance ϵ on the observed data to 0.0025, which means we stop optimizing B with SVT when the relative error is less than ϵ or the maximum number of iterations is reached. When using SVT to optimize B, we set the relative error $error_{SVT}$ in the following format:(34)$$\begin{array}{c} error_{SVT}=\frac{∥P_{\Omega }\left(T-B^{\left(k\right)}M^{T}\right){∥}_{F}^{2}}{∥P_{\Omega }{\left(T\right)∥}_{F}^{2}}. \end{array}$$

For the estimated tensor X obtained at each iteration in Algorithm 4, the relative error is assessed as (Equation 32) and (Equation 33), for other methods considered, the relative errors are also assessed using the same form. In addition to the relative error and the maximum number of iterations, Algorithm 4 also terminates if the difference of the relative errors obtained from the algorithm for two consecutive steps is less than a certain threshold η, i.e.,
(35)$$\begin{array}{c} \left|\frac{error_{iter}^{\left(k+1\right)}-error_{iter}^{\left(k\right)}}{error_{iter}^{\left(k\right)}}\right|<\eta , \end{array}$$
where the $error_{iter}^{\left(k\right)}$ represents the relative error obtained at the *k*th iteration.

The initial rank is set to $R^{\left(0\right)}=\left[r,r,r\right],\;r=5,10,15$. Table 1 and Table 2 respectively show recovery error (Equation 32) and (Equation 33) of test problem 1. It can be seen that when the given initial rank does not match the ground truth data, gHOI, M${}^{2}$SA and Tucker–Wopt have large errors, while our RATC-TD has small errors. We emphasize that our proposed method does not require a given tensor rank; the initial rank of our proposed algorithm is set to the size of the tensor we aim to complete, i.e., $R^{\left(0\right)}=$ size(T).

In this experiment, we use our RATC-TD algorithm to estimate the tensor ranks. We next use the M${}^{2}$SA method to improve the estimated results. Our algorithm can be considered an initial step for other algorithms, and the effects of tensor completion are also sound when only our method is used for optimization. The recovery results without using M${}^{2}$SA are shown in Table 3. As seen from Table 2, since our method was used for optimization in advance, compared with the results obtained by simply using the M${}^{2}$SA method with a given rank, it gives a better recovery effect. Notably, our proposed algorithm can also stably complete the tensor under a small sampling rate and does not need to give an appropriate rank in advance.

### 5.2. Test Problem 2: The Recovery of Fourth-Order Tensors

Similarly to test problem 1, following the procedures in [9], we in this test problem generate the object tensor $T\in R^{30\times 30\times 30\times 30}$ by multiplying a randomly generated kernel $G\in R^{5\times 5\times 5\times 5}$ with randomly generated factor matrices of size $R^{30\times 5}$. The ground truth data is defined by T with Gaussian noise added (the same setting as in test problem 1). For gHOI, M${}^{2}$SA, Tucker–Wopt and our proposed RATC-TD, different initial ranks are tested, and the sampling rate *r* is set to 0.1. The corresponding number of entries in the observed data set Ω is r|T|, and the index of each entry in Ω is randomly generated through the uniform distribution. The errors for this test problem are shown in Table 4, where $error_{obs}$ is the error on the observed data set (Equation 32) and $error_{val}$ is the error on the validation set (Equation 33). It can be seen that our RATC-TD method performs well for this test problem.

### 5.3. Test Problem 3: The Recovery of Real Missing Pictures

In this test problem, real practical data are considered, and the performance of our RATC-TD is compared with that of gHOI and M${}^{2}$SA. The initial complete images considered are shown in Figure 1. The data format of each image can be regarded as a third-order tensor. Each image here is stored as a tensor with size 256×256×3, where the third dimension is 3, representing three color channels of red, green, and blue. The ground truth data T for this test problem are the images in Figure 1 (details are as follows).

Two ways to construct partial images are considered, and our method is tested to recover the original images using the partial images. First, some black lines are added to the images, which can be considered a kind of structural missing, and the corrupted images are shown in the first column of Figure 2. The black line parts of images correspond to $\bar{\Omega }$, and the rest correspond to Ω. Second, after the initial image is converted into the data format of a third-order tensor T, the observed index set Ω is constructed with each index randomly generated through the uniform distribution, and the size of Ω is r|T| (where the sampling rate is r=0.1). The images associated with Ω are shown in the first column of Figure 3.

From Figure 2, it is clear that the images obtained by our RATC-TD are closer to the ground truth images than the ones obtained by gHOI and M${}^{2}$SA. The numerical results representing the qualities of recovery are shown in Table 5. In addition to the errors $error_{obs}$ (Equation 32) on the observation set and the errors $error_{val}$ (Equation 33) on the unobserved data set, we also use SSIM (structure similarity index measure) [29] and PSNR (peak signal-to-noise ratio) parameters to evaluate the effect of our image restoration, which are often used to assess the quality of image restoration. SSIM ranges from 0 to 1; the larger the SSIM is, the smaller the image distortion is. Similarly, the larger the PSNR is, the less distortion there is. PSNR is used to measure the difference between two images. For the restored image X and the original image T, the PSNR between them is given by the following formula:(36)$$\begin{array}{c} \mathrm{PSNR}=10\times \mathrm{lg}\left(\frac{{\mathrm{Maxpixel}}^{2}}{\mathrm{MSE}}\right), \end{array}$$
where Maxpixel is the maximum value of the image pixel. In our test problem, Maxpixel=255, and MSE is the mean square error between X and T. For the case with sampling rate r=0.1, the corresponding restoration results are shown in Figure 3, and the numerical results representing the qualities of recovery are shown in Table 6. In this case, we only use 10 percent of the ground truth data, but it can be seen that our RATC-TD gives effective recovery results for this test problem.

For the gHOI method and the M${}^{2}$SA method, which need to be given the rank of the initial tensor in advance, we choose the initial rank as $R^{\left(0\right)}=\left[20,20,3\right]$. It is worth noting that for real image data, it is difficult for us to know the exact Tucker rank in advance. When there is no way to know the rank in advance, in order to use gHOI and M${}^{2}$SA to achieve data completion, the initial rank can only be constantly adjusted through experiments, or the rank is given based on prior experience.

## 6. Conclusions

In this paper, we propose a novel rank-adaptive tensor completion based on the Tucker decomposition (RATC-TD) method. As the existing tensor completion methods based on Tucker decomposition, such as gHOI, $M^{2}\mathrm{SA}$ and Tucker–Wopt, typically require the tensor rank given a priori, overestimating or underestimating the tensor ranks can lead to poor results. Inspired by the RaHOOI algorithm, we propose our algorithm based on the HOOI structure. Our proposed algorithm can adaptively estimate the multilinear rank of data in the process of tensor completion. We show the algorithm’s effectiveness through experiments on completing synthetic data and genuine pictures. The results of our algorithm can also provide effective initial data for other tensor completion methods, such as $M^{2}\mathrm{SA}$. The further work we expect is to extend the algorithm to higher-dimensional problems, which requires us to optimize the algorithm further to reduce the algorithm’s computational time.

## Figures and Tables

**Figure 1 entropy-25-00225-f001:**
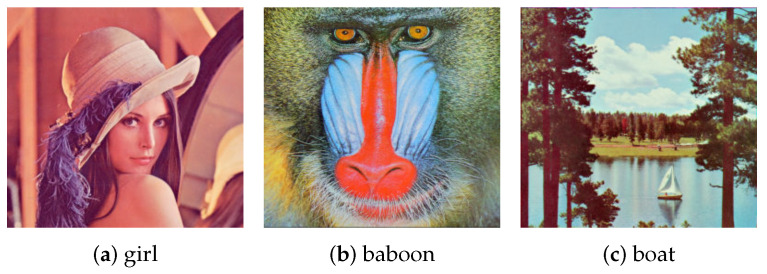
The original images, which are referred to as `girl’, `baboon’ and `boat’, test problem 3.

**Figure 2 entropy-25-00225-f002:**
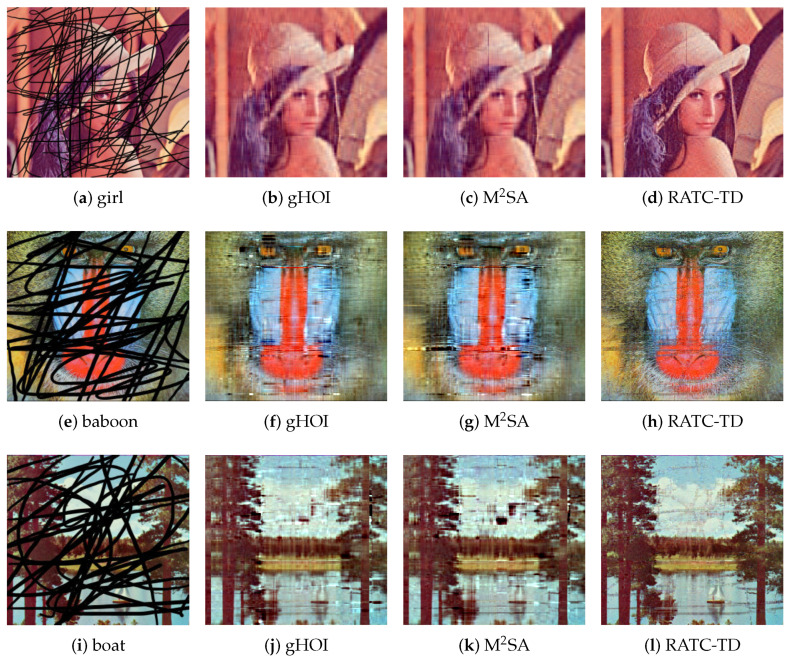
The restoration results of the images with black lines, test problem 3.

**Figure 3 entropy-25-00225-f003:**
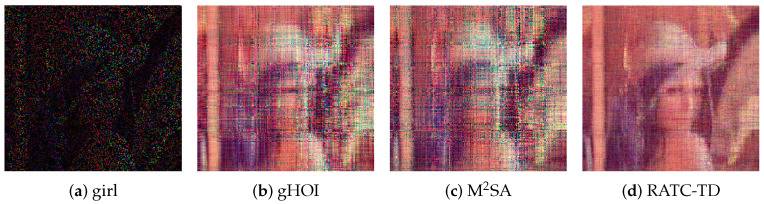
The restoration results with sampling rate r=0.1, test problem 3.

**Table 1 entropy-25-00225-t001:** Relative error (Equation 32) comparison with different initial rank on observed data, test problem 1.

Sampling Ratio	Initial Rank	M${}^{2}$SA	gHOI	Tucker–Wopt	RATC-TD
0.05	5, 5, 5	0.0354	0.0431	0.0354	0.0354
0.05	10, 10, 10	0.0322	0.0404	0.0284	0.0354
0.05	15, 15, 15	0.0270	0.0362	0.1128	0.0354
0.1	5, 5, 5	0.0264	0.0278	0.0264	0.0264
0.1	10, 10, 10	0.0250	0.0267	0.0233	0.0264
0.1	15, 15, 15	0.0229	0.0229	0.0199	0.0264
0.2	5, 5, 5	0.0235	0.0236	0.0235	0.0235
0.2	10, 10, 10	0.0230	0.0229	0.0225	0.0235
0.2	15, 15, 15	0.0223	0.0215	0.0208	0.0235

**Table 2 entropy-25-00225-t002:** Relative error (Equation 33) comparison with different initial rank on unknown data, test problem 1.

Sampling Ratio	Initial Rank	M${}^{2}$SA	gHOI	Tucker–Wopt	RATC-TD
0.05	5, 5, 5	0.0187	0.0518	0.0187	0.0186
0.05	10, 10, 10	0.3438	0.4192	0.0429	0.0186
0.05	15, 15, 15	0.6262	0.6143	1.4444	0.0186
0.1	5, 5, 5	0.0153	0.0231	0.0153	0.0153
0.1	10, 10, 10	0.2300	0.3102	0.0241	0.0153
0.1	15, 15, 15	0.4935	0.4521	0.0610	0.0153
0.2	5, 5, 5	0.0145	0.0162	0.0145	0.0145
0.2	10, 10, 10	0.2480	0.2390	0.0181	0.0145
0.2	15, 15, 15	0.4872	0.2800	0.0222	0.0145

**Table 3 entropy-25-00225-t003:** Numerical results only using our proposed algorithm RATC-TD, test problem 1.

Sampling Ratio	${\mathrm{error}}_{\mathrm{obs}}$	${\mathrm{error}}_{\mathrm{val}}$
0.05	0.0500	0.0410
0.1	0.0489	0.0523
0.2	0.0495	0.0443

**Table 4 entropy-25-00225-t004:** The results of tensor completion quality of methods in the case of sampling rate *r* = 0.1, test problem 2.

Types of Error	Initial Rank	M${}^{2}$SA	gHOI	Tucker–Wopt	RATC-TD
$error_{obs}$	5, 5, 5, 5	0.0300	0.0342	0.0300	0.0300
$error_{val}$	5, 5, 5, 5	0.0147	0.0251	0.0147	0.0147
$error_{obs}$	15, 15, 15, 15	0.0258	0.0255	0.1552	0.0300
$error_{val}$	15, 15, 15, 15	0.7130	0.4807	6.6025	0.0147

**Table 5 entropy-25-00225-t005:** Numerical characterization of the recovery quality of images with black lines, test problem 3.

**Girl**	**PSNR**	**SSIM**	${\mathrm{error}}_{\mathrm{obs}}$	${\mathrm{error}}_{\mathrm{val}}$
missing figure	11.4347	0.5044	/	/
gHOI	24.8558	0.9536	0.0236	0.0985
M${}^{2}$SA	24.8196	0.9532	0.0235	0.1080
RATC-TD	28.9981	0.9833	0.0099	0.0751
**Baboon**	**PSNR**	**SSIM**	${\mathrm{error}}_{\mathrm{obs}}$	${\mathrm{error}}_{\mathrm{val}}$
missing figure	8.6372	0.1878	/	/
gHOI	18.3661	0.7477	0.0268	0.1670
M${}^{2}$SA	17.2923	0.6948	0.0263	0.2341
RATC-TD	20.9857	0.8612	0.0098	0.1004
**Boat**	**PSNR**	**SSIM**	${\mathrm{error}}_{\mathrm{obs}}$	${\mathrm{error}}_{\mathrm{val}}$
missing figure	8.5502	0.3229	/	/
gHOI	17.4365	0.8717	0.0282	0.2419
M${}^{2}$SA	16.9671	0.8556	0.0276	0.2635
RATC-TD	22.0162	0.9524	0.0099	0.0998

**Table 6 entropy-25-00225-t006:** Numerical characterization of the recovery quality of images obtained by random sampling at sampling rate of 0.1, test problem 3.

**Girl**	**PSNR**	**SSIM**	${\mathrm{error}}_{\mathrm{obs}}$	${\mathrm{error}}_{\mathrm{val}}$
missing figure	5.5747	0.0329	/	/
gHOI	13.2045	0.6585	0.0309	0.1336
M${}^{2}$SA	11.9943	0.5977	0.0305	0.1663
RATC-TD	18.9566	0.8049	0.0100	0.0764
**Baboon**	**PSNR**	**SSIM**	${\mathrm{error}}_{\mathrm{obs}}$	${\mathrm{error}}_{\mathrm{val}}$
missing figure	5.8165	0.0306	/	/
gHOI	13.6954	0.5862	0.0319	0.1121
M${}^{2}$SA	12.8840	0.5404	0.0351	0.1441
RATC-TD	17.4101	0.7008	0.0098	0.0968
**Boat**	**PSNR**	**SSIM**	${\mathrm{error}}_{\mathrm{obs}}$	${\mathrm{error}}_{\mathrm{val}}$
missing figure	5.6110	0.0346	/	/
gHOI	12.5033	0.7168	0.0312	0.1336
M${}^{2}$SA	11.6136	0.6549	0.0343	0.1615
RATC-TD	17.4527	0.8735	0.0096	0.0869

## Data Availability

Not applicable.

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
