# Peer review of "Rank-Adaptive Tensor Completion Based on Tucker Decomposition"

_entropy, 2023, doi:10.3390/e25020225_

Round 1

Reviewer 1 Report

The paper describes an interesting approach on tensor completion. Although it presents valuable practical results, the theoretical contributions are presented in a somewhat fuzzy manner. The authors should try to elaborate more on the theory presented in section 4.

Some other issues should also be addressed. Examples:
=> Consider replacing some expressions with more formal/clear language -> "are in urgent need" or "is directly up to", "but", etc.
=> "This section is to summarize" -> "This section summarizes"
=> Consider merging subsection 4.3 into 4.2, or expanding it in order to properly emphasize the author's contributions. It seems to be too short and somewhat forced.
=> Please consider introducing the abbreviation "RATC-TD" when it first appears.
=> Maybe the first paragraph of sub-section 5.1 should be part of the theoretical content presented in the previous section.
=> Please check on some missing commas.

Reviewer 2 Report

Pleasee see the attachment. 
